# Infrastructure Shaming and Consequences for Management of Urban WEF Security Nexus in China and India

**Daphne Gondhalekar *** and **Jörg E. Drewes**

Chair of Urban Water Systems Engineering, Department of Civil, Geo and Environmental Engineering, Technical University of Munich (TUM), 80333 München, Germany; jdrewes@tum.de
*   Correspondence: d.gondhalekar@tum.de

**Abstract:** Worldwide, consumption of resources such as water, energy and food continues to rise exponentially despite environmental and climatic change related challenges. Centralized sewerage systems continue to be implemented worldwide despite being very water and energy intensive, and although this is not always the best option for regions facing water scarcity. Deploying the Water-Energy-Food (WEF) Nexus approach, particularly through alternative technology options that can support decentralized water reclamation with integrated resource recovery, can enable resource conservation and more effective management of the WEF security Nexus for local governments with limited capacities. However, a certain pattern of "business as usual" infrastructure development and investment linked to infrastructure shaming continuously reinforces implementation of centralized sewerage systems, thereby hampering deployment of alternative technology options. This study uses two typical case study towns, Shaxi in China and Leh in India, to describe this pattern. The study finds that alternative technology approaches were in place in both towns. Yet after international consulting companies got involved, centralized sewerage systems were implemented despite limited water availability and large segments of the population not having flush toilets. This study discusses management of the WEF security Nexus implications thereof in the context of cities worldwide and a systemic socio-technical transition to a circular economy.

**Keywords:** climate change; natural resources conservation; urban Water-Energy-Food (WEF) security Nexus; water reclamation with integrated resource recovery; China; India

## 1. Introduction

With ongoing urbanization, globalization and industrialization, the growth-dependent global economy is driving over-consumption of natural resources such as water, energy and food worldwide, especially in cities [1]. This has led to degradation of the environment, a process that is interacting with and further accelerated by a dangerous alteration to the climate [2]. Despite the apparent risks associated with climate change, resource demand continues to rise worldwide. Global water demand is expected to exceed supply by 55% by 2050 [3]. Under the current policy scenario, energy demand is projected to increase by 1.3% per year to 2040, with concomitant rise in energy-related emissions [4]. Food demand is also expected to increase by 60% by 2050 due to population growth and increased consumption. Agriculture already consumes 70% of total water, and its energy consumption is expected to increase by 84% by 2050, which also affects $CO_2$ emissions [5]. At the same time, a high level of consumption of these resources remains the ideal of a good quality of life.

Scientists have been warning of the consequences of over-exploitation of natural resources and questioning the long-term generation of positive growth rates for half a century [6]. The founding fathers of the free market economy had also imagined a different development: they assumed that when wealth had increased to a certain level enabling everyone to cover important needs, the economy would be sated and stop growing (Keynes, 1930, in [7]). Instead, satisfaction with quality of life in countries like Germany stopped

growing in the 1970s [7] and on-going economic growth has served 0.7% of the world population in amassing 45.2% of global wealth [8] whilst billions of people continue to live in dire poverty and lack basic services. Although economists are pointing out the negative consequences of unequal income distribution [9,10], research and policy tend to focus on how to continue pushing economic growth whilst procuring ever more natural resources rather than curbing resource demand.

To enable achievement of the United Nations Sustainable Development Goals (SDGs), ideals and policies urgently need to evolve. Many cities have limited knowledge, human and financial resources to address complex challenges systemically. Hence, they urgently require innovative and practicable solutions that deliver high-impact outcomes that leverage on synergies between climate change mitigation and adaptation approaches. Solutions also need to address social, economic and environmental issues, and be accompanied by facilitating methods and tools. Integrated urban planning solutions enabling resource conservation and recovery are urgently needed as a basis for a circular economy.

The Water-Energy-Food (WEF) Nexus approach [11] can support development of such solutions. The approach highlights interlinkages between the water, energy and food sectors, such as that it takes much energy to supply freshwater and remove and treat wastewater, or that much water is needed to produce energy and food [12]. Characterizing such interrelationships can help to target synergies between these sectors and help to avoid tensions [13]. For example, it was found that water stocks returns of listed companies operating in the water industry are sensitive to agriculture and energy price trends [14]. This demonstrates the existence of a WEF Nexus in financial terms that was particularly exacerbated during the 2008 financial crisis [14]. The approach underlines responsible governance [15] and that a new perception of water is needed, with the water-food link being of the highest social and political significance [12]. Particularly, water management and security are key to sustainable social and economic development and to effective management of the WEF security Nexus.

Existing urban water-related infrastructure systems are essentially linear and not conducive to implementation of the WEF Nexus approach. For example, the dominant "business as usual" centralized sewerage systems modeled on Western cities are very water and energy intensive. To cover an entire city area and flush sewage to treatment plants usually located at the urban periphery, a large volume of water needs to be supplied and consumed to prevent a huge piping network from clogging. Further, centralized sewerage systems are expensive to operate and maintain. Despite these factors, the concept of a flush toilet coupled to a centralized sewerage system remains the preferred option, seen as a symbol of "modernity", even though it has been termed "ecologically mindless" [16].

Water reclamation with integrated resource recovery is a key entry point for cities to leverage on the WEF Nexus approach. The process can enable recovery of (1) water for various uses including irrigation of agriculture and parks, toilet flushing, groundwater recharge, car-washing and others particularly in water-scarce regions, (2) energy, e.g., thermal energy as well as energy generation, e.g., biogas for cooking or conversion to electricity and (3) nutrients, e.g., in the form of organic fertilizer, and substances like bio-plastics, minerals like iron, salts and antibiotics. These resources have increasing economic value in currently emerging markets, supporting operationalization. Thus, water reclamation with integrated resource recovery can support effective management of the WEF security Nexus.

In centralized sewerage systems, resource recovery is encumbered as sewage is diluted through the large amounts of water used. Resource recovery in Western cities is very far from standard practice. Globally, only 11.6% of treated wastewater is directly reused [17]. A significant part of wastewater treatment plants' energy demand can be covered locally with recovered energy. However, in the US there are 104 wastewater treatment plants using biogas to produce a total of 90 MW (US EPA, 2011, in [18]). With 16,000 publicly owned wastewater treatment plants in the US serving 75% of the total population [19], this is only a tiny fraction. In Germany, only few wastewater treatment plants use energy regained for their operation: in 2016, 1/8th of treatment plants produced 1450 GWh of

electricity from biogas gained from sewage treatment, 90% of which was used to run the treatment plants [20].

Wastewater is a huge untapped resource [21]. To tap it effectively, decentralized wastewater management systems are gaining popularity worldwide. They can support water conservation as they are smaller and need less water to flush, and hence allow for more effective recovery of nutrients and energy through less dilution. Investment costs are 20–50%, and operation and maintenance costs 5–25% of conventional treatment plants [21]. A total of 90% of the investment cost of centralized sewerage systems, due to their sheer size, is usually for conveyance systems. However, although many alternative wastewater treatment technology options exist that can be deployed in decentralized wastewater management systems, these are not readily available because they have seldom been tested at urban scales. In the meantime, centralized sewerage systems continue to be implemented worldwide, although in many countries they may no longer be the most viable option [21].

This "technology bottleneck" is caused by a certain pattern of wastewater management infrastructure development and investment reinforcing continued implementation of centralized sewerage systems by default, which directly hampers implementation of technology alternatives and thereby also negatively impacts management capacity of the WEF security Nexus. To support a systemic socio-technical transition and overcome this bottleneck we first need to understand how communities conceive, develop, plan and implement wastewater management infrastructure.

The aim of this study was to illustrate challenges to implementing water reclamation with integrated resource recovery through the analysis of the multi-stakeholder decision-making process surrounding wastewater management infrastructure development and investment in the context of towns facing large-scale development pressures in China and India. In particular, the role of international consulting companies in this process is investigated. As this is an under-researched topic, the study thereby makes an original contribution to this research gap.

In the structure of this paper, various factors influencing water security as a basis for the WEF security Nexus are analyzed whilst highlighting interlinkages to energy and food securities using two case study towns in the results section, before drawing inferences from their juxtaposition in the discussion section in terms of implications for management of the WEF security Nexus. In the conclusion, preliminary key elements for enablement of decentralized water reclamation with integrated resource recovery are deduced.

## 2. Materials and Methods

This study takes a case study approach. This empirical method is considered suitable for the detailed explorative study of phenomena that are as yet little studied, complex, contextual, that must be analyzed using multiple sources of evidence, and which particularly defy use of quantitative methods [22,23]. Two typical towns, Shaxi in the state of Yunnan in China, and Leh in Ladakh, a former region of the State of Jammu and Kashmir that recently became a Union Territory in India, are taken as case studies. These cases are typical because both towns experienced very rapid economic growth due to tourism industry development, and face serious water availability, pollution and wastewater management challenges as a result. Hence, they had to develop a new wastewater management approach within a short period of time. These cases are further juxtaposed because they have certain similar and different characteristics: both are small towns in relatively remote locations of limited knowledge, human and financial resources, but in quite different socio-cultural settings. In both towns, development of an alternative technology approach was well underway that could have enabled water reclamation with integrated resource recovery, but which was abandoned in favor of the implementation of a centralize sewerage system.

In order to investigate the reasons behind this development, Shaxi and Leh are each analyzed using four key water-related factors influencing water security as a basis for the WEF security Nexus, namely: (1) economic development; (2) water pollution issues—resulting

from (1); (3) policy and governance geared towards resolving (2); and (4) infrastructure development and investment.

Members of the Nexus@TUM Research Group have worked in Shaxi in 2011–2012 and in Leh since 2012. The main method of analysis in both case studies are semi-structured interviews. During seven days in October 2011, a total of ten semi-structured interviews with representatives of local government at town and county levels, local environmental NGOs and local households were conducted in Shaxi. The interviews are based on E. Economy's survey in which a political science approach is taken to investigate the role of political institutions and politics in shaping China's environmental and development pathway [24]. It included questions such as: Who are the key actors and what is their relative power? How are resources allocated to environmental protection? How is environmental policy formulated and implemented? What incentives exist for government, business, and society to advance goals of environmental protection, specifically water reclamation with integrated resource recovery? In a similar line of questioning, in Leh 20 semi-structured interviews were conducted in 2012–2018 also with representatives of the local government, local environmental NGOs and local households. In Leh, questionnaire surveys of 200 households and 400 hotels and guesthouses were undertaken.

For both case studies, documentation review was also employed as a methodology to analyze technical consultants' reports concerning the development of wastewater management systems. To be able to assess factors linked to poverty and economic development, statistical national census data was also reviewed.

## 3. Results

### 3.1. Shaxi, China

The town of Shaxi is situated in a remote valley in Jianchuan County in the Dali Bai Autonomous Prefecture, Yunnan Province, in Southwest China. It is about 50 km south of the Tibetan High Plateau in the Himalayan foothills [25]. Administratively, Shaxi has a central town with 2300 inhabitants [26] and 12 villages [25]. In 2009, Shaxi had a total population of 22,000 inhabitants [27], which was forecast to reach 34,000 by 2020 [25] due to rapid growth of the tourism industry.

At the start of this century, tourists were still virtually unknown in Shaxi. In 2001, however, it was discovered that Shaxi, located on the southern Silk Road, has several ancient artefacts of international interest including a theatre, temple and caravanserai. Numbers of visitors have increased exponentially since then. A new highway completed in 2014 connecting Lijiang, the nearest international airport 120 km away with Dali the prefectural capital passes close by Shaxi, enabling fast access. Thus, Shaxi is facing large-scale development pressures. In 2009, 90% of the population of Shaxi were still farmers [27] and with its historic marketplace and intact cultural landscape of paddy field terraces and natural rivers, Shaxi is very picturesque [25].

The Shaxi Rehabilitation Project was incepted in 2001 as a public–private partnership between the Swiss Federal Institute of Technology Zurich (ETH) and the People's Government of Jianchuan County to protect and preserve this natural and cultural heritage [28]. In China, it is considered a very successful and unique cultural rehabilitation project [29] and includes a plan for ecological sanitation developed by the Swiss Federal Institute of Environmental Science and Technology (EAWAG).

In China, human feces are commonly used as organic agricultural fertilizer. This has prompted discussion on whether decentralized wastewater management systems may be better suited to the Chinese context than centralized ones in order support use of this resource.

### 3.1.1. Economic Development

Yunnan Province is one of the poorest provinces of China. The World Bank set the benchmark for absolute poverty in 1990 at less than 1 $ purchasing power parity (PPP). In 2004, this was approximately 1.08 $ per day for China [30]. The Chinese benchmark for

absolute poverty is lower at 625 ¥/year [30], which was 75 US$ per year or 0.2 US$ per day in 2004. Of Yunnan's rural population, 2.86 million people live in absolute poverty according to the Chinese standard and 7.64 million are low-income [30]. Despite substantial socio-economic development in Yunnan in recent decades, this has impacted mainly larger urban centers, whilst rural areas still lagging strongly behind [31], partially due to adverse weather conditions and geographical isolation [30]. About 80% of the population of Shaxi are of the ethnic minority Bai [32].

In Shaxi, the average annual net income is ca. 1000 ¥ or 120 US$ per capita, considered very low even by Chinese standards [27]. Farmers interviewed said that rice sells for only 4 ¥/kg. They receive an annual subsidy of 100 ¥/year. With such low income, many farmers in Shaxi struggle just to meet the cost of living and average expenditures are about 10% higher than average earnings [27]. Primary and middle school is free in China but high school costs 1000 ¥ per year. Shaxi has no high school; thus, children have to go to Jianchuan or Dali, where room, board and uniform cost about 7000–8000 ¥ per year. State health insurance in China, officially mandatory, costs 40 ¥ per year. To this, the State contributes 20 ¥ and reimburses ca. 50% of medical fees. Thus, the financial burden of health coverage on a farmer is quite high.

The tourism industry has developed very rapidly in Shaxi. Tourism was of little significance before the Shaxi Rehabilitation Project was initiated, but by 2003 there were 6000 visitors per year [28] and 28,000 in 2011. However, these tourists usually stay only a few hours in Shaxi and less than 30% stay overnight [33]. In the past decade, there has been a significant increase in guesthouses, but only about 100 people in Shaxi work in restaurants, hotels and tourism-related retail [27]. Further, most tourists come on package tours with transport and guiding pre-arranged. So far, economic growth only marginally impacted standard of living of the local community [33], as most economic gain goes to foreign investors.

Nonetheless, tourism is rapidly pushing up prices and farmers are increasingly leaving Shaxi in search of additional income: 75% of families have one member working outside Shaxi Valley, that is almost all male inhabitants aged between 20 and 40 [27], to earn 1000–2000 ¥/month in industries including construction, retail, etc. Thus, to afford high school for one child, one family member must work outside Shaxi for several months per year.

The growth of the tourism industry in Shaxi has put significant pressure on the local government to come up with a wastewater management solution. Although tourists only stay for a few hours, adequate sanitary infrastructure needs to be provided to support Shaxi's attractiveness as a tourist destination.

### 3.1.2. Water Pollution Issues

Available water resources in Shaxi are estimated by the Jianchuan County Water Department to be 200 million $m^3$ per year, of which only 10% are currently being used. However, Shaxi faces serious water pollution concerns: the Heihui River running through Shaxi Valley, which was clean and abundant in fish a few decades ago, was found to have alarming levels of water pollution and in 2013 according to the local government, national-level funding was approved to mitigate pollution.

As agriculture next to tourism is the only relevant industry in Shaxi, it is assumed that there are only three sources of water pollution: feces, chemicals and solid waste. In Shaxi, in accordance with national guidelines fecal sludge is commonly used to fertilize rice paddy and cornfields twice a year, approximately two months after the planting of the rice and immediately after the harvest. For this purpose, it is stored for at least one month, after which it becomes liquid, which is easier to distribute. Fecal sludge is also used for vegetable fields, for which it is only stored for a few days but then mixed with farm manure and buried under at least half a meter of soil onto which vegetables are planted. Farmers use households' private pit latrines, animal stables and school and public toilets to collect fecal sludge. EAWAG found that 55% of households had private pit latrines and in addition there were 15 public toilets, mostly in very poor sanitary condition, and

thus assumed a public health risk due to unhygienic conditions [32]. Further, EAWAG estimated that 300–600 tons of feces are spread onto fields in Shaxi annually and that 50% of the effluents from latrines infiltrate into the soil and groundwater, corresponding to 500–1000 tons annually [32].

Increased use of chemicals in households in the form of soap and detergents also contributes to water pollution in Shaxi [32]. Further, despite use of fecal sludge as organic agricultural fertilizer, over-use of chemical fertilizers prompted the start of a soil sampling campaign in Shaxi in 2008. However, the local government estimates that it will take 5–10 years to change farmers' practices. One farmer stated that he uses 50 kg of chemical fertilizer on his land, in addition to 3 tons of fecal sludge annually.

Further, EAWAG estimated that 400–900 tons of solid waste are being produced in Shaxi annually [32], about half of which is collected by a private company and landfilled. The rest is fed to animals, burned or dumped into canals or the river. Glass, metal and paper are only partially recycled, but a central recycling factory for Shaxi is planned.

### 3.1.3. Policy and Governance

In Shaxi, water management is divided into the three main components irrigation, drinking water provision and sanitation. In 1954, the first irrigation channel was constructed at Shaxi to divert water from the Heihui River at the top of the valley into the rice paddy fields and, in the decades following, the channel system was improved and elaborated. Prior to the 1980s, irrigation was not being managed in a comprehensive manner in Shaxi. In 1991, the Jianchuan County Government mandated the Town-level Shaxi Water Resources Station, responsible for managing water resources and irrigation including infrastructure such as channels and pumps, mountain spring pipes and reservoirs. Each April/May, village water committee representatives meet to agree on a water distribution plan. The county level approves the plan before integration into the national plan, and provides funds for water infrastructure. Groundwater is not used for irrigation but only by households on vegetable patches.

Before the 1980s and until about 2003, groundwater wells were used for drinking water. Today, drinking water is piped from mountain springs and wells are no longer used for drinking water due to groundwater pollution. In Shaxi Valley, the price of drinking water varies depending on pipe ownership: the central town government pipe water is most expensive at 0.8 ¥/m$^3$. For villages owning their pipe, drinking water is free. Others pay 5–10 ¥ per family per year. There are no government subsidies for drinking water provision at the household level. Construction started in 1992 to provide every household in Shaxi with piped drinking water and was completed in 2003. The government constructed the main pipes below the road. However, the government provides no household level subsidy for piped water connections and private investment is needed at about 10 ¥/m of piping. The Shaxi central town mountain water pipe has now reached its full carrying capacity and a new one is being planned. Sanitation in Shaxi is the responsibility of a separate Town-Level Sanitation Office. However, before 1992 there was no sanitation planning, and there is still no comprehensive plan.

Over 80% of Jianchuan County is forest. Logging for construction material and firewood on the slopes of the hills in Jianchuan County and ensuing diminished ability of the hills to retain water has direct impact on lower-lying reaches of the Yangtze. The disastrous large-scale flooding of the Yangtze River in 1998 prompted the national re-forestation policy of 2000. Since then, de-forestation in northwest Yunnan is strictly prohibited. For forest protection in Jianchuan, the government subsidizes energy alternatives to burning wood: electricity is heavily subsidized at 30% of the usual cost and most households have electric stoves. Further, 40–50% of households have solar water heaters after a subsidy was introduced in 2008. Biogas generation also exists.

Due to its location in the upper reaches of the Yangtze River, Jianchuan County thus has strong decision-making power in terms of environmental protection and decided for instance in 2003 to ban the development of industries. Implementation of such projects in

China much depends on the vision of the local leader. Local leaders are appointed every five years with a maximum period of ten years and can get promoted to a higher administrative level, but only if they fulfil economic goals in line with the current Five-Year Plan.

### 3.1.4. Infrastructure Development and Investment

At the time of the EAWAG survey in 2002, pit latrines were the only sanitation infrastructure, only 30–40% of households had individual taps, the quality of groundwater was not monitored in Shaxi and thus diarrhea linked to unhygienic conditions was identified as a serious public health issue [32]. As farmers proved very interested in improved sanitation conditions but were unwilling to relinquish fecal sludge as valuable organic agricultural fertilizer, EAWAG recommended implementing a decentralized sanitation system at the household level in Shaxi using urine separation toilets with septic tanks [32]. This system aimed to protect groundwater resources and sanitize fecal sludge by storing it in airtight conditions in a septic tank for at least one month [32]. These septic tanks were to be implemented in each household, underneath the courtyard.

Following the EAWAG recommendations, 10 pilot urine diversion toilets were implemented with government subsidy in 2003–2004, as well as a simplified sewer system. However, these pilot toilets are not in use because locals found them not to be very user-friendly and flushing is considered more convenient. In 2009, the Jianchuan County Government built a central treatment plant in Shaxi, which treats sludge from the septic tanks and channels the effluent into the Heihui River.

However, on a larger scale, the EAWAG proposal was not implemented, for several reasons. Despite acknowledging that the Chinese national guidelines of 1987 advise storing fecal sludge for at least one month before use, the EAWAG report states that farmers tend to empty the chambers of the latrines whenever they need fertilizer, regardless of the retention time in the chamber and that fresh feces are spread on the fields onto crops, thus considerably increasing the risk of infectious disease transmission [32]. In our survey, respondents claimed that the national guidelines are being followed, because it is well known that spreading fresh feces onto food crops is a considerable public health risk. According to our survey, fertilizer is commonly only applied to fields only before the sowing period, or in the case of vegetable crops, buried in pits with the vegetables planted on top of a soil layer, as described above.

Further, although EAWAG noted that culturally local people always construct toilets outside their courtyards [32], they recommended septic tanks be constructed below the courtyards. This our survey found to be equally culturally unacceptable. As the courtyard floor is used to prepare and dry agricultural products, having septage being collected below is thought disgusting. The land outside the courtyards does not belong to the households. Having a small toilet house as an appendage seems socially acceptable, but not digging up an area of land able to hold a septic tank. Having the septic tank outside the courtyards would have facilitated sanitized fecal sludge access. This would have necessitated a dialogue with the local government on relinquishing land rights for sanitation infrastructure development, which did not take place.

Almost none of the households and less than 30% of guesthouses have septic tanks. To build a septic tank in Shaxi costs around 10,000 ¥, but no government subsidies exist for sanitation improvement at the household level, e.g., to upgrade toilets or implement septic tanks. The cost of 200 ¥ per dry toilet, deemed affordable for the local population [32], is clearly too expensive at 20% of the average annual income, not to mention the cost of a septic tank. The local government refused to fund the proposed decentralized sanitation system. A key selling point of EAWAG was the sanitized fecal sludge. However, this according to the national guidelines, was already available. Perhaps thus the sense of implementing septic tanks in each household, on top of feeling miffed by the formulations in the EAWAG report and the plan not having been properly thought through, eluded the local government.

Still pressed, however, to implement a solution to the groundwater pollution issue, in 2011, the provincial government announced the decision to fund a centralized sewerage system in Shaxi, to be implemented by a private company.

### 3.2. Leh, India

Leh Town, the capital of Ladakh in India, is located in a remote semi-arid region in the Himalayas in the Indus River Valley at an altitude of 3500 m above sea level. Leh is considered one of the fastest-expanding small towns in India [34]. Ladakh has been governed by the Ladakh Autonomous Hill Development Council (LAHDC) since 2005. According to the 2011 Census of India, conducted every 10 years, Leh Town has a population of approximately 30,000. In addition, several tens of thousands of army personnel and migrant workers live in Leh.

Adjoining a dense historical town center, Leh's urban area is spread throughout a green valley of agricultural fields lined by trees and watered by a dense network of streams fed by glacial and snow melt water, surrounded by a desert landscape. This intricate cultural landscape is the product of hundreds of years of very careful management of limited water resources. Leh was a traditional agricultural irrigation society until only a few decades ago [35] where water was used extremely sparingly. The traditional sanitation system, the Ladakhi dry toilet, does not require any water and is still used by about 30% of households as a source of organic agricultural fertilizer. However, although Leh is situated in a desert, the rapid growth of the tourism industry in Leh is pushing up the consumption of water and concomitant wastewater production exponentially. Until only a few decades ago, Leh was almost self-sustained in terms of food. Since then, around 30% of agricultural fields have fallen barren or were built up, and almost all food is imported, at further energy cost. This pushes up the price particularly of fresh vegetables, which in winter are not sufficiently available for the local population. Hence, serious questions of water, energy and food security present themselves in Leh that need to be addressed urgently. These could further be exacerbated with climate change impacting water availability, e.g., through glacier retreat.

### 3.2.1. Economic Development

Being in proximity to the border with Pakistan and China, Leh was opened for tourism only in 1974. Since then, the number of visitors to Leh has increased exponentially, especially in the last decade. The Leh Tourism Department estimates that around 400,000 tourists visit Leh annually. As winters are harsh, the vast majority of these tourists visit Leh in summer between April and October. In order to cater to these tourists, there has been a huge increase in hotels and guesthouses in Leh. In the 1980s, there were only 24 hotels and guesthouses, but by 1990 there were 62, by 2000 there were 117, by 2010 there were 282, and just from 2010 to 2012, the number increased to 360 in business, with another 60 not yet in business or under construction [36].

These tourist accommodations are increasingly building en-suite bathrooms with flush toilets and showers to be able to set higher room prices. Despite the rapid growth of the tourism industry, in Leh the median household income per year was found to be only INR 150,000 (~1800 EUR) in 2013. As an indicator of relative poverty, 7% of households were found to have health insurance, of which 20% were privately covered. A total of 99% of tourists prefer to use a flush toilet rather than a Ladakhi dry toilet [37] and most also prefer to use a shower rather than a water bucket to bathe. In Leh, 90% of the freshwater supply is obtained from groundwater sources originating from glacial melt.

### 3.2.2. Water Pollution Issues

To cater to the huge increase in water demand, the local government supplies freshwater through a centralized and a decentralized system [38]. Groundwater is extracted via tubewells from the Indus river aquifer. From there, it is lifted via 4 lifting stations 200–300 m vertically and several kilometers horizontally to Leh's urban area. In addition,

the Public Health Engineering (PHE) Department of the local government supplies water from the aquifer underneath Leh. This water is fed to reservoirs distributed around Leh and then distributed by a gravity pipe system with several hundred public and private water taps and by water tankers. In the summer months, the supplied amount is 3000–4000 $m^3/d$, of which 1000–2000 $m^3/d$ are extracted via tubewells from the Indus river aquifer, 1300 $m^3/d$ extracted from tubewells tapping the Leh aquifer and 800 $m^3/d$ channeled from various springs near the top of Leh. This public water supply system in Leh is very energy intensive: it is estimated to require 600 L of diesel daily [39] but manages to supply water for only 2–3 h per day.

According to our survey, the public water supply is considered insufficient to run a hotel or guesthouse with showers and flush toilets. Thus, by 2015, 60% of hotels and guesthouses had built private borewells as a source of decentralized water supply, up from 42% in 2009 (Akhtar, 2010). Individual hotels and guesthouses extract several thousand liters of groundwater daily in the tourist season. For example, a hotel owner interviewed of a hotel with 18 en-suite rooms reported extracting up to 8000 liters per day during the tourist season. Groundwater extraction is not regulated in Leh, and hence there is concern about its over-abstraction. Further, the main source of energy in Leh is hydropower, but electricity supply is not stable and Leh faces regular power outages. Therefore, many hotels and guesthouses own diesel power generators to bridge power supply gaps and sustain use of the borewells.

Leh does not have a systematic wastewater management system in place. The increase in tourist accommodations with their concomitant wastewater production poses a public health risk. Hotels and guesthouses dispose of wastewater mainly through so-called septic tanks which in actual fact are soak pits that are not properly managed so that wastewater seeps into the ground. Further, 33% of borewells in Leh are too close to septic tanks and soak pits according to World Health Organization guidelines (WHO, 1996). As a result, the Leh aquifer has become polluted by wastewater seepage and action is urgently required to protect the groundwater quality as a source of drinking water [40].

In terms of the local population's perception of freshwater pollution, our questionnaire survey also revealed that although 98% of households thought that drinking water quality is safe in Leh, 49% of households thought drinking water quality today is worse than 10 years ago. A total of 35% of households reported having problems with their drinking water in terms of smell, taste or color. Lack of adequate sanitation system, i.e., soak pits, were thought by 31% of households to be the main source for groundwater pollution. Increased use of chemical fertilizer in agriculture was also perceived as a water quality threat. A total of 40% of households thought drinking water pollution is related to diarrhea. Thus, drinking water pollution is a serious concern for the local population.

### 3.2.3. Policy and Governance

The Ladakh 2025 Vision Document [41] was formulated in 2005 as a vehicle to discuss the development of the region. It covers the traditional and new economy, and physical and social infrastructure, with water resources discussed under physical infrastructure. The document points out that Ladakh's traditional land-based economy has significantly declined in recent decades due to the new abundance of economic opportunities arising with the growth of the tourism industry amongst others, but that it should be sustained to avoid excessive dependence on the outside world for such critical resources as food, and hence make Ladakh more resilient to crisis situations. Hence, land-based economy should be preserved and encouraged. In a semi-arid region, it is especially crucial to conserve water resources and protect these from pollution. Energy conservation through use of renewable energy is also advocated. The document calls for "moral fortitude to withstand the vicissitudes of an uncertain age" and summarizes the vision as follows: "By 2025, Ladakh will emerge as the country's best model of hill area development in a challenging environment, with its sustainability embedded in ecological protection, cultural heritage

and human development", where "Ladakh will be seen as an ideal society geared towards economic self-reliance, full employment and enhanced quality of life for its people."

In terms of water, the Vision Document discusses alternative forms of irrigation and the possibility of a redistribution of land holdings to enable new irrigation projects. The creation of new artificial water bodies in the form of artificial glaciers and reservoirs is underlined to support conservation of water resources, potentially accompanied by an institutional change in form of a reorganization of administrative departments supported by inputs from a research entity. Water supply in 2005 was 1500 $m^3$/d, and demand in 2025 estimated at 8000 $m^3$/d. It advocates that around 1100 $m^3$/d of this demand in 2025 could potentially be supplied by tapping wastewater as a resource. The Vision Document acknowledges an "indisputable decline" in the amount of water resources available in Ladakh, due to decrease in snow cover, glacial area and precipitation figures. Finally, the document advocates more efficient water use and underlines that water and wastewater conservation needs more adequate policy support in Ladakh.

In terms of water quality, the document recommends, amongst others, to stop dirty water and effluents from directly entering water resources, particularly those used for drinking water, set standards and norms for the kinds of discharges allowed into water bodies, and enforced through proper legislation if required, preserve traditional systems ensuring sanctity of water sources, and to encourage conservative use of water, including water saving systems for toilets, showers, etc., especially in guesthouses and hotels, and to plant crops that require less water. The Vision Document acknowledges that active participation of the local population is crucial for the implementation of such points. For the future, it recommends to systematically assess water resources in the region including groundwater and establish a regulatory body for apportioning groundwater rights, and states that a strategy is needed including wastewater storage as well as groundwater recharge. The Vision Document further advocates levying a "green tax" on all visitors to Ladakh as a progressive measure and means of raising revenues to reduce environmental impact.

Hence, as early as 2005, most key ideas for sustainable water management were well in place in Leh. However, implementation is complicated by various factors: LAHDC lacked capacity to take the water reclamation with resource recovery element of the Vision Document forwards. A small team of engineers of PHE is facing a daily challenge to keep the water supply system with its lifting stations running. Further, LAHDC until very recently was a semi-autonomous local government with little funding to implement innovative projects. Finally, the construction season in Leh is very short, from May to October, which coincides with the agricultural season. In the winters there is time to plan, but many Ladakhis leave for lower altitude locations as Leh is too cold.

At the same time, due to the exponential rise in numbers of visitors to Leh and its increasing visibility as an international tourist destination, pressure to act on the wastewater management situation was steadily building up on LAHDC.

### 3.2.4. Infrastructure Development and Investment

In 2008, LAHDC commissioned an international consulting company to develop a detailed project report to fix the wastewater management situation in Leh [42]. The consulting company carried out various field studies, engineering surveys and investigations to collect the necessary data to prepare the Detailed Project Report (DPR), and designed a centralized sewerage system in accordance with the Indian national government guidelines [43] for the year 2040, using 2010 as the base year.

In the DPR, sewage is collected from Leh's urban wards via a gravity- and pump-based piping system at a central wastewater treatment plant (WWTP) situated at Stok village at the base of Leh near the River Indus. Due to Leh's challenging topography, the sewer needs to be laid at depths varying from 2 m to 8 m below ground to avoid freezing in winter, when temperatures in Leh reach −30 °C. The system comprises a 20 km trunk and

36 km branch sewer, 21 km gravity mains and 400 m rising mains, that is around 80 km of piping to cover the area of Leh town of about 19 km$^2$.

To design the size specifications of the WWTP, the DPR estimated the population of Leh in the year 2040 as follows: it used Indian National Census data from the last eight decades (in 1921 Leh had 2401 inhabitants, and 28.639 in 2001), and five different conventional methods to project the population figure with these respective results: (1) Arithmetical Progression Method (41.431); (2) Geometrical Increase Method (75.891); (3) State Urban Average Method (82.251); (4) Incremental Increase Method (67.679); and (5) Graphical Method (72.191). Since these resulting figures diverged quite widely, the consulting company used their average as the projected population of Leh in 2040, namely 67.888. The floating population of Leh was further estimated at 150% of the resident population of which 20% were considered to be present in Leh at any given time. With this, the total projected population of Leh in the year 2040 was estimated at 94.365.

In India according to national guidelines, in urban areas with a centralized sewerage system the local government has to provide 135 liters per capita per day (lpcd) of potable water, of which 35 lpcd is needed just to flush the system [43]. However, in urban areas without a centralized sewerage system, 75 lpcd is sufficient. In Leh, PHE aims to provide 75 lpcd but according to our survey, the local population may be consuming as little as 21 lpcd. Thus, with the implementation of the centralized sewerage system, LAHDC is urging the local population to consume around six times as much water as before. This, to many Ladakhis who have been using water extremely sparingly for centuries and are very much aware that they live in a desert, seems preposterous.

Of the 135 lpcd to be supplied, also as per the national guidelines, 80% is taken to be the sewage flow generated. Hence, the WWTP was designed for an average capacity of 13,000 m$^3$/d and a peak discharge of 33,000 m$^3$/d in 2040. For the sewage treatment, a conventional extended aeration process was recommended. The WWTP is planned at the foot of Leh, where wastewater is to be treated, reused locally for urban agriculture irrigation and the rest discharged to the Indus River.

The DPR further envisaged that only households would be connected to the centralized sewerage system. Hotels and guesthouses, although these produce approximately 1000 m$^3$/d of wastewater in summer, a vast resource that is currently not being utilized, are not planned to be connected. The consulting company undertook a household survey to determine the annual sewerage services charge to cover the operation and maintenance costs of the system, and an estimated 1000 Indian Rupees (INR) per household was deemed acceptable. Relative to the median annual income in Leh, however, this represents a significant sum. Further, as mentioned above, according to our survey in 2013, across Leh 60% of households do not have a flush toilet. In summer, 67% of households use Ladakhi dry toilets and 28% use a combination of traditional Ladakhi dry toilets and flush toilets. In contrast, only 1% of tourists use a Ladakhi dry toilet [37]. In winter, 91% of households use a Ladakhi dry toilet and only 6% a flush toilet, as these tend to freeze in winter. The cost of connecting to the centralized sewerage system and the necessary sanitary infrastructure to consume 135 lpcd of water is to be borne by the households. The DPR assumes that all households will connect and pay the service charge, but for those that do not have flush toilets or showers this may make little sense. Further, in the DPR, hotels and guesthouses are not planned to connect, although these produce substantial amounts of wastewater, because they are mandated to build on-site treatment systems under the national Zero Liquid Discharge Act, which applies to industries. Hence, there is a fair degree of uncertainty about whether the amount of wastewater estimated by the DPR will in fact reach the WWTP, what revenues will be available to run the system and what will happen to the system in winter, when also large parts of the public water supply system are frozen.

Despite such drawbacks in the system design, and lacking alternatives, LAHDC representatives started lobbying the Indian Central Government for funding to implement the centralized sewerage system. Interviews with local government representatives revealed that energy availability is seen as the main bottleneck to its operation as water resources

available through the Indus river aquifer are considered ample. Ladakh has leased huge areas of land to the national government for solar energy development, which is expected to supply Leh more reliably with energy. In 2014, the Indian Central Government granted LAHDC the funds to implement the DPR.

During the ensuing implementation, various challenges surfaced: so far, about 50% of the sewerage system has been constructed, but the funding estimate of 2009 is no longer sufficient to construct the system as designed in the DPR. Hence, some pumps needed to be cut, leaving about 50% of Leh unconnected and needing an alternative solution. Further, the design of the WWTP did not demonstrate a disinfection step for effluent treatment before agricultural reuse. Hence, due to pollution concerns, there have been public protests to its implementation in Stok, the village at the foot of Leh planned to host the WWTP. As a result, construction of the WWTP has not yet commenced.

Out of necessity, hotels and guesthouses are taking a different approach. The Grand Dragon Hotel in Leh was the first to install an on-site solution in 2015: using a moving bed bio-reactor (MBBR) wastewater is aerated, then clarified in a settling tank, filtered through sand and finally disinfected with UV irradiation. The quality of the effluent conforms to safety standards and is used to irrigate the lawn in the hotel garden. This system successfully reclaims water, but is not viable for most hotels in Leh—it has to be underground to maintain the correct operating temperature, which roughly doubles capital expenditure. The aeration requires much energy and the various system components need constant monitoring by two full-time in-house engineers, entailing high operation and maintenance costs. The system must also be kept running in winter to keep the bacteria in the MBBR alive, so that sufficient toilets must be flushed by the staff even when there are few guests.

As an alternative for other hotels and guesthouses, a fecal sludge treatment plant (FSTP) was constructed on the outskirts of Leh in 2016. Sludge is collected from septic tanks and soak pits by vacuum truck, and transported to the FSTP where it is emptied into a series of planted drying beds. The dried sludge is used as organic fertilizer. The effluent is filtered using sand and stored in an open tank for superficial UV disinfection, and can be used to irrigate agricultural crops. Hence, this system is effective for resource recovery. However, the sludge must be pumped out and transported long distances to the FSTP, which is energy intensive in terms of diesel consumption. Further, as mentioned previously, most so-called septic tanks in Leh are actually soak pits that are not sealed at the bottom. Hence wastewater seeps into the ground, leaving tanks void of sludge that can be extracted. Many of the septic tanks are also difficult to access as they have never been emptied and are covered with soil. Finally, many roads in Leh are too narrow for the truck to be able to get to all hotels and guesthouses.

In order to provide the 135 lpcd required to flush the centralized system, LAHDC is currently also constructing a new water supply pipeline to supplement the existing system and lift more water to Leh from the Indus river aquifer. This new pipeline is estimated to further push up energy demand in terms of diesel significantly [39]. Further, in a country such as India, and in such a large system, a relatively high percentage of leakages in the water supply and sewage collection systems must also be assumed. In Leh, leakages are further exacerbated by the uneven topography making pipe laying difficult, and harsh winters causing pipe bursts. Hence, the centralized sewerage system probably, on top of its other issues, once and if it is completed, cannot be effective in fulfilling its original purpose: to conserve groundwater quality by curbing sewage seepage. Unless of course no sewage ever finds its way into it.

For more information on the case study in Leh, see the documentary film: "If not now, when? Planning for the urban Water-Energy-Food Nexus": https://vimeo.com/142941443, password: wefnexusleh.

## 4. Discussion

In this section, implications for management of WEF security Nexus of the results of the process of infrastructure development and investment described above are discussed. Particularly, in terms of implementation potential of decentralized water reclamation with integrated resource recovery and the role of foreign intervention in enabling the implementation of technology options that are alternatives to "business as usual" centralized sewerage systems.

The Shaxi case shows us that although water management issues are complex, interlinkages between the sectors water, energy and food and implications for a WEF security Nexus can be clearly characterized. Safe drinking water access may become an issue in Shaxi in the future as the mountain spring supplying Shaxi has reached its limit. With population and tourism industry growth, Shaxi may have to reconsider groundwater as a source of drinking water in the future. Thus, groundwater quality needs to be protected. Implementation of a centralized sewerage system will require far more water from mountain springs or groundwater sources to be procured and flushed. Supplying more water needing to be treated after use further has energy implications. The cost of flush toilets will fall to the local population, who have little incentive to invest in these and connect to the system. Hygiene is an important motivation for the local population and persistence of pit latrines will continue to negatively impact groundwater quality. For the majority of the population in Shaxi, for whom the priority is to exit the poverty trap via investment into education and health, sanitation infrastructure investment is not affordable and not considered a priority. The centralized sewerage system will make fecal sludge less readily available as a source of organic fertilizer and drive increased use of chemical fertilizers and thus groundwater pollution, with implications for food security.

A decentralized system enabling water reclamation with integrated resource recovery on the other hand would require less water, hence also less energy, could even be used to generate biogas as a well-established form of renewable energy in China at the household level, and preserve fecal sludge for use in agriculture to support groundwater quality conservation. Tourists represent a large untapped resource for poverty alleviation. Under current circumstances, the local population does not profit directly from tourism. An option may be to levy a tourism tax to subsidize sanitation infrastructure investment, e.g., septic tanks, and raise the standard of living of local farmers to help preserve Shaxi's cultural landscape, which will otherwise eventually disappear.

Despite a complex system of governance, if local leaders are convinced, they have the power to implement alternative solutions in Shaxi. What does it take to convince them? EAWAG was in a good position to propose an alternative technology option that might have been a viable solution. However, that solution was not properly designed taking all relevant local factors into account. The result was the creation of a low-trust situation in EAWAG, with a pressing problem still needing to be solved. In that knowledge and capacity vacuum, the local government, not having any other options, decided to implement the default, the centralized sewerage system.

In Leh, the situation is similar in that benefits of water reclamation with integrated resource recovery for WEF security Nexus enhancement are easily highlighted. However, again implementation is challenging on several fronts. In Leh, the availability of water is still being questioned. Although the Indian Government Central Groundwater Board (CGWB) has conducted several studies to estimate availability of groundwater in Leh using satellite imagery and geophysical methods [44], estimates are only available for the district level. At the town level it is very difficult, with a multi-layer aquifer in a glacial moraine, to estimate and we will probably never know the capacity of the Leh aquifer. Still, it seems starkly obvious that in a desert setting, under climate change conditions and where glacial melt is the only source of freshwater, this needs to be conserved. In Leh, there is concern that restricting groundwater extraction could harm tourism industry development and growth.

Yet as in Shaxi, the local government did not have capacity to develop an alternative to the centralized sewerage system. We have been lobbying the local government in Leh since 2012 for decentralized water reclamation with integrated resource recovery. At that time, the consulting company plan had already been formulated, with its shortcomings. For example, the population estimate is derived by dubious methods to say the least. LAHDC has been much aware that the system will run into issues during construction, operation and maintenance, and that the STP seems to be over-dimensioned.

The TUM team on the other hand could not formulate an engineered plan for an alternative technology option in the space of time until funding was granted. Despite involvement of organizations and NGOs across India working on decentralized water reclamation with integrated resource recovery, because of its relatively unique high-altitude and semi-arid situation, existing options could not be translated easily to Leh.

LAHDC made efforts to have certain elements of the DPR redesigned. In practice however, the consulting company exited the scene after the DPR was completed. In its place stepped another consulting company which offered to redesign the DPR for 10% of the project value. This LAHDC could not afford. Worried that funds from Delhi might dry up if not spent, there was no choice but to start implementing the DPR as the first consulting company had designed it. As soon as funds for the centralized sewerage system were granted, construction started.

Decentralized water reclamation with integrated resource recovery also in Leh can support water conservation, as such systems require much shorter pipes and therefore less water to flush. Hence, in the pockets not being able to connect to the centralized sewerage system the water supply could stay at 75 lpcd, still conform to national guidelines and take pressure off the water supply system. Water can further be reused locally, e.g., for agricultural irrigation instead of having to be lifted from the WWTP to fields in the Leh town area. Lifting and treating less water also implies energy conservation. Further, less diluted sewage from decentralized systems can support energy recovery or generation, e.g., biogas. Again, such decentralized systems can also support conservation of fecal sludge as a source of organic fertilizer, a traditional practice that will otherwise decline.

*Synthesis*

In both Shaxi and Leh, despite being in quite different socio-cultural settings, the starting issues were similar as was the final outcome. In both places awareness that water resources are limited and hence water conservation is needed is high whilst facing huge development pressures driving up water consumption. Further, in both places a very pragmatic approach is taken to fecal sludge as a valuable resource. As a result, approaches were developed which pushed for the implementation of decentralized wastewater management systems enabling water conservation and reclamation with integrated resource recovery. At the same time, alternative technology options that could have been deployed in these decentralized systems were not properly translated into the respective local contexts so that the solutions could have worked. In the case of Shaxi, the septic tanks proposed by EAWAG were not integrated properly into the social and urban fabric and were too expensive for the local population. In Leh, alternative technology options like constructed wetlands remain to be trialed and tested for their effectiveness under the particular climatic conditions. Even if the capacity of the Leh aquifer is not known, it is very energy intensive to provide water, and water conservation can be part of a climate change mitigation and adaptation strategy. In both Shaxi and Leh, there was an ensuing knowledge and capacity vacuum following these attempts to develop alternative technology options. In this vacuum, the "business as usual" centralized sewerage system was sold to local governments as the "solution that works" to solve the urgent wastewater management issue on hand, despite the evident constraints. That is even when it seems quite apparent to the population and decisionmakers that, because of a lack of water, lack of people who own or can afford to invest in sanitation infrastructure such as flush toilets, etc., this may not be the best option.

## 5. Conclusions

In Shaxi and Leh, despite huge development pressures, a certain critical mass of elements supports potential implementation of water reclamation with integrated resource recovery as a key to enhance management of the WEF security Nexus. This is based on a certain level of awareness of water scarcity and availability issues having been achieved that sustains momentum for action. The local governments, despite limited capacities, have leaders able to formulate a vision and take decisions based on it. In both localities, support by independent research and other organizations like NGOs fertilizes the potential for innovation in terms of water reclamation with integrated resource recovery which can enhance water conservation and security.

However, although many technology alternatives to centralized sewerage systems are available that can enable water reclamation with integrated resource recovery at urban scales, this study has demonstrated that even when water stress and awareness are high as in Shaxi and Leh, it is not a straight-forward process to implement these. After international consulting companies got involved in the process of infrastructure development and investment, centralized sewerage systems were implemented in both towns despite their largely lacking the sanitary infrastructure required to operate such systems, e.g., flush toilets. As the centralized sewerage systems require additional water and energy for their operation, these systems are impacting water, energy and related food securities negatively, exacerbating environmental issues and undermining the capacity of the local governments in Shaxi and Leh to manage the WEF security Nexus and achieve the SDGs. This study concludes that these international consulting companies represent larger-scale driving forces that are continuously reinforcing a pattern of implementation of centralized sewerage systems as the "business as usual" technology option to manage wastewater.

This study further surmises that these key implications for the management of the WEF security Nexus are not particular to Shaxi and Leh. The PM of India said in discussing India's Smart City Mission, that "smart" is not necessarily linked to information and communication technology but in the first line has to be "something that works" to address existing challenges. Addressing water-related challenges is the key element of this mission, as a crucial foundation for sustainable urban and economic development. Such smart solutions to water-related challenges need to be tailored to the local context and align with local interests. Implementing alternative technology options enabling water reclamation with integrated resource recovery is one such smart solution that can readily be tailored to local contexts. However, worldwide, towns like Shaxi and Leh continue to fall prey to international consulting companies who are interested in selling large-scale centralized sewerage systems that are expensive, resource intensive, complex to operate and maintain and often come with service contracts attached.

Wastewater management remains in many localities a sensitive topic. A "space of shame" is created when local governments are put under pressure to address inadequate wastewater management despite lacking knowledge, human and financial capacities to do so effectively. This opens the door to international consulting companies' designs. In the discourse on wastewater management, hygiene issues are used as a catalyzer for improvement. Yet, in the development of wastewater management infrastructure, the focus lies on procuring large amounts of water and energy to flush away waste, and not on quality of life improvement, the apparent pledge of economic growth.

At the start of this paper, we postulated that the growth-dependent global economy needs questioning, particularly in terms of natural resources over-consumption patterns. This study has demonstrated how, at a local scale, a few actors profit from infrastructure investment, whilst the majority of the population are left to face the detrimental effect on the environment. Still, this pattern of infrastructure development and investment is firmly rooted in the idea of economic growth linked to increased natural resources consumption. Currently, if natural resources are conserved, very few people profit directly in monetary terms. However, individuals representing companies in turn representing the 0.7% amassing the lion's share of global wealth do profit directly from the implementation

of resource intensive infrastructures, and ensuing consumption of large amounts of water and energy. Addressing the concomitant environmental destruction is also a very profitable business. As Noam Chomsky writes, our economic system has intrinsic institutional properties driving environmental destruction and paying no attention to externalities [45]. If we seriously want to change the process of infrastructure development and investment described in this study to enable natural resources conservation, we will need to collectively rebel against mechanisms, institutions and entities driving profit-seeking behavior.

*5.1. Recommendations*

This study advocates that the integration of the following steps into the infrastructure development and investment process can support enablement of water reclamation with integrated resource recovery:

- Inclusive dialogue without "finger-pointing" focusing on value of water and wastewater
- multi-stakeholder participatory processes to support co-design of solutions
- research linked to piloting to make alternative technology options readily available
- stronger integration of SDGs, and climate change mitigation and adaptation goals into policy and governance contexts
- incentives for local leaders, e.g., indicator-based point system for ecological in addition to economic performance, linked to re-election prospects
- incentives at household level, e.g., subsidies for sanitation infrastructure; and
- independent funding opportunities for local governments.

With the support of such an inclusive process, small towns like Shaxi and Leh, where generally alternative technology options may be more easily implemented than in large cities, can act as lighthouse examples of decentralized water reclamation with integrated resource recovery and more effective management of the WEF security Nexus.

*5.2. Limitations*

The case study approach entails certain limitations. Obviously, two cases is a very small number. The results of this paper do not allow for results to be systematically generalized. Exploration of more cases is needed in order to build a base for a systematic comparative analysis to be conducted, also including a larger number of cases to cover strategically a larger range of similarities and differences, hence enabling a systematic deduction of results.

*5.3. Future Research*

To build a base for a systematic comparative analysis to be conducted on the topic of this paper is the object of future research. Further, the topic begs to be expanded to include important research questions with policy implications such as: how much of the national debts of countries like China and India is due to infrastructure development? Which strings attached to loans by multilateral funding agencies serve to perpetuate implementation of water and energy intensive infrastructures which drive high natural resources consumption patterns? Where are the turnkeys in this system from an international environmental and climate political perspective that can enable achievement of the SDGs? What are the alternatives to economic growth? How can a more circular economy be enabled that can support natural resources conservation?

The evolution to a more circular economy through a systemic socio-technical transition can build on cases of enabling environments, as in Shaxi and Leh, which are already in place. It is the aim of future increasingly action-oriented research to empower local leaders with vision to make out of many small infrastructure investment opportunities a big investment in a sustainable future as a new model of economic growth, and treat wastewater as the asset it is, thereby augmenting local management capacity for the WEF security Nexus. A critical mass of such cases will help to tip the balance for cities worldwide.

**Author Contributions:** For this study, D.G. was responsible for the funding acquisition, project administration, study conceptualization, survey implementation, analysis, writing the original draft, and J.E.D. contributed to the writing review and editing, and project supervision. All authors have read and agreed to the published version of the manuscript.

**Funding:** This research is supported by a Marie Curie International Reintegration Grant within the 7th European Community Framework Programme (PIRG06-GA-2009-256555) and the German Research Foundation (DFG) (KE 1710/1-1).

**Institutional Review Board Statement:** Not applicable.

**Informed Consent Statement:** Not applicable.

**Data Availability Statement:** Not applicable.

**Acknowledgments:** We thank our research partners in Shaxi, China, the Shaxi Rehabilitation Project and LEP Consultants, for facilitating the implementation of the interview survey in Shaxi, and our research partner in Leh, Ladakh Ecological Development Group (LEDeG), India, for organizing and carrying out data collection.

**Conflicts of Interest:** The authors declare no conflict of interest.

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
