# Peer review of "Infrastructure Shaming and Consequences for Management of Urban WEF Security Nexus in China and India"

_water, doi:10.3390/w13030267_

Round 1

Reviewer 1 Report

Thank you for the opportunity to read this manuscript. Below some suggestions:

In the section introduction, the Authors should better explain the aim of the study and emphasize the originality of the study

The section "Material and methods" is poor. The authors must improve this section better explaining the approach adopted for the case study analysis. Interesting sources and good examples to improve this section are:

Galati, A., Crescimanno, M., Vrontis, D., & Siggia, D. (2020). Contribution to the sustainability challenges of the food-delivery sector: Finding from the deliveroo italy case study. Sustainability, 12(17), 7045.

Fiore, M., Galati, A., Gołębiewski, J., & Drejerska, N. (2020). Stakeholders' involvement in establishing sustainable business models. British Food Journal.

Giacomarra, M., Crescimanno, M., Sakka, G., & Galati, A. (2019). Stakeholder engagement toward value co-creation in the F&B packaging industry. EuroMed Journal of Business.

The questionnaire for the interview have a scientific foundation?

Results are clearly presented and discussed

The end section should be improved emphasizing the main implications of the study, the limitations related to the case study approach, and future research

Author Response

  • Thank you for the opportunity to read this manuscript. Below some suggestions: In the section introduction, the Authors should better explain the aim of the study and emphasize the originality of the study
  • RESPONSE: Thank you, yes this has been improved with sentences added in various places to make these clearer.
  • The section "Material and methods" is poor. The authors must improve this section better explaining the approach adopted for the case study analysis. Interesting sources and good examples to improve this section are: Galati, A., Crescimanno, M., Vrontis, D., & Siggia, D. (2020). Contribution to the sustainability challenges of the food-delivery sector: Finding from the deliveroo italy case study. Sustainability, 12(17), 7045; Fiore, M., Galati, A., Gołębiewski, J., & Drejerska, N. (2020). Stakeholders' involvement in establishing sustainable business models. British Food Journal; Giacomarra, M., Crescimanno, M., Sakka, G., & Galati, A. (2019). Stakeholder engagement toward value co-creation in the F&B packaging industry. EuroMed Journal of Business.
  • RESPONSE: Great, this is very helpful indeed. The section has been updated accordingly as well as adding more detailed information.
  • The questionnaire for the interview have a scientific foundation?
  • RESPONSE: Yes, the approach of the study of E. Economy which is referred to in the paper and which the survey is modeled on is defined to illustrate the scientific foundation.
  • Results are clearly presented and discussed.
  • RESPONSE: Thank you
  • The end section should be improved emphasizing the main implications of the study, the limitations related to the case study approach, and future research.
  • RESPONSE: The section has been updated accordingly.

Reviewer 2 Report

This paper deals with the study of the infrastructure shaming and consequences for management of urban WEF Security Nexus. Authors highlight a certain pattern of “business as usual” infrastructure development and investment linked to infrastructure shaming and use two typical case study towns, Shaxi in China and Leh in India, to describe this pattern.

This topic appears original and very interesting and seems relevant for readers of  WATER. The goal of the study is well justified in the text. The title clearly describes the article and the abstract reflects its content.

However, I think the content of the article could be slightly improved if authors consider the following aspects:

P1, L. 32-39: please update, if possible, the data

p.2/3: in the Introduction, when reporting on water sector issues, I would suggest considering the work of “Vandone et al. (2018). The impact of energy and agriculture prices on the stock performance of the water industry. Water resources and economics, 23, 14-27)” in which there is a particular view of funding for the water sector.

p.2 Line 59-66: When inserting the topic of nexus I think it is very important to mention the work that started the idea of nexus itself, i.e.:

Bazilian et al. (2011). Considering the energy, water and food nexus: Towards an integrated modelling approach. Energy policy, 39(12), 7896-7906.

p. 14: in the Conclusion I suggest reading the paper: “Peri et al. (2017). Volatility spillover between water, energy and food. Sustainability, 9(6), 1071” that gives a completely different view of the nexus, because it measures WFE relationships in financial terms. I suggest to consider also this point of view.

Author Response

  • This paper deals with the study of the infrastructure shaming and consequences for management of urban WEF Security Nexus. Authors highlight a certain pattern of “business as usual” infrastructure development and investment linked to infrastructure shaming and use two typical case study towns, Shaxi in China and Leh in India, to describe this pattern.This topic appears original and very interesting and seems relevant for readers of  WATER. The goal of the study is well justified in the text. The title clearly describes the article and the abstract reflects its content.
  • RESPONSE: Thank you
  • However, I think the content of the article could be slightly improved if authors consider the following aspects: P1, L. 32-39: please update, if possible, the data
  • RESPONSE: Yes, these sources were rather outdated and have been replaced with more recent references.
  • 2/3: in the Introduction, when reporting on water sector issues, I would suggest considering the work of “Vandone et al. (2018). The impact of energy and agriculture prices on the stock performance of the water industry. Water resources and economics, 23, 14-27)” in which there is a particular view of funding for the water sector.
  • RESPONSE: Thank you, this has been added as a reference.
  • 2 Line 59-66: When inserting the topic of nexus I think it is very important to mention the work that started the idea of nexus itself, i.e.: Bazilian et al. (2011). Considering the energy, water and food nexus: Towards an integrated modelling approach. Energy policy, 39(12), 7896-7906.
  • RESPONSE: Yes, absolutely correct, this has been added as a reference.
  • 14: in the Conclusion I suggest reading the paper: “Peri et al. (2017). Volatility spillover between water, energy and food. Sustainability, 9(6), 1071” that gives a completely different view of the nexus, because it measures WFE relationships in financial terms. I suggest to consider also this point of view.
  • RESPONSE: Thank you, since we aimed not to put any references in the conclusion with the exception of Chomsky, this point has been added on page 2.

Reviewer 3 Report

This paper has low soundness and might not be very interesting to readers.

Some parts of the introduction should be transferred to the methods with more details.

The methods lack details 

The conclusions seem to be recommendations . The author may divide it into two parts.

Author Response

  • This paper has low soundness and might not be very interesting to readers.
  • RESPONSE: Thank you, comments on how to improve the soundness would be much appreciated.
  • Some parts of the introduction should be transferred to the methods with more details.
  • RESPONSE: Yes, this is true, the section has been updated accordingly.
  • The methods lack details.
  • RESPONSE: More details have been added to this section.
  • The conclusions seem to be recommendations . The author may divide it into two parts.
  • RESPONSE: The section has been updated accordingly.